# Unsupervised Speech Recognition

**Alexei Baevski**$^{\triangle}$  **Wei-Ning Hsu**$^{\triangle}$  **Alexis Conneau**$^{\square *}$  **Michael Auli**$^{\triangle}$

$^{\triangle}$ Facebook AI  $^{\square}$ Google AI

## Abstract

Despite rapid progress in the recent past, current speech recognition systems still require labeled training data which limits this technology to a small fraction of the languages spoken around the globe. This paper describes wav2vec-U, short for wav2vec Unsupervised, a method to train speech recognition models without any labeled data. We leverage self-supervised speech representations to segment unlabeled audio and learn a mapping from these representations to phonemes via adversarial training. The right representations are key to the success of our method. Compared to the best previous unsupervised work, wav2vec-U reduces the phone error rate on the TIMIT benchmark from 26.1 to 11.3. On the larger English Librispeech benchmark, wav2vec-U achieves a word error rate of 5.9 on test-other, rivaling some of the best published systems trained on 960 hours of labeled data from only two years ago. We also experiment on nine other languages, including low-resource languages such as Kyrgyz, Swahili and Tatar. The code is available at `https://github.com/pytorch/fairseq/tree/master/examples/wav2vec/unsupervised`

## 1   Introduction

Speech recognition performance on the much studied English Librispeech benchmark [Panayotov et al., 2015] has seen rapid improvement over the last few years due to advances in model architectures [Dong et al., 2018, Synnaeve et al., 2020, Gulati et al., 2020], semi-supervised learning [Xu et al., 2020b, Park et al., 2020] and self-supervised learning [van den Oord et al., 2018, Chung and Glass, 2018, Chung et al., 2019b, Baevski et al., 2020c]. However, all of these techniques require transcribed speech data which is not available for the vast majority of the nearly 7,000 languages of the world [Lewis et al., 2016]. As a result, speech recognition technology is only available for about 125 different languages [Google, 2021]. On the other hand, humans learn a lot about speech simply by listening to others around them and without explicit supervision [Werker and Tees, 1984, Hirsh-Pasek et al., 1987, Polka and Werker, 1994, Jusczyk et al., 1999, Johnson and Jusczyk, 2001].

Unsupervised learning has been very successful in machine translation resulting in systems that obtain remarkable accuracy given no labeled training data at all [Conneau et al., 2018, Lample et al., 2018, Artetxe et al., 2018]. Inspired by this, there has been some work on unsupervised speech recognition based on learning to align unlabeled text and audio [Yeh et al., 2019] or adversarial learning [Liu et al., 2018, Chen et al., 2019]. These approaches showed promising initial results but their error rates are still high, with evaluation being limited to the small-scale and clean TIMIT benchmark.

In this work, we introduce a framework for unsupervised learning of speech recognition models. Wav2vec-U, or wav2vec Unsupervised, leverages self-supervised representations from wav2vec 2.0 [Baevski et al., 2020c] to embed the speech audio and to segment the audio into units with a simple k-means clustering method (see Figure 1 for an illustration of our approach). We find that the

---

$^{*}$Work done while at Facebook AI.

35th Conference on Neural Information Processing Systems (NeurIPS 2021).

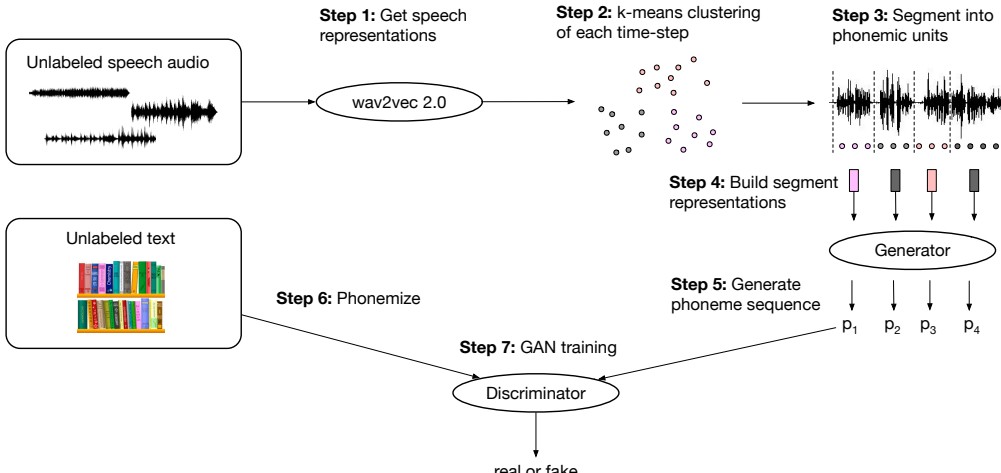

Figure 1: Illustration of wav2vec Unsupervised: we learn self-supervised representations with wav2vec 2.0 on unlabeled speech audio (Step 1), identify clusters in the representations with k-means (Step 2) to segment the audio (Step 3). Next, we build segment representations by mean pooling the wav2vec 2.0 representations, performing PCA and a second mean pooling step between adjacent segments (Step 4). This is input to the generator which outputs a phoneme sequence (Step 5) fed to the discriminator, similar to phonemized unlabeled text (Step 6), for adversarial training (Step 7).

quality of the audio representations is key to the success of unsupervised speech recognition. Similar to Liu et al. [2018] and Chen et al. [2019], we learn a mapping between segments and phonemes using adversarial training but different to their work, we also enable the algorithm to label segments as silences. We also introduce an unsupervised cross-validation metric to enable model development without labeled development data. Our unsupervised speech recognition model, the generator, is very lightweight: it consists of a single temporal convolution comprising only about 90k parameters to which we input frozen wav2vec 2.0 representations.

Experimental results demonstrate the viability of the framework for a variety of settings and languages. wav2vec-U improves the phone error rate (PER) on the small-scale TIMIT benchmark from 26.1 to 11.3 compared to the next best known unsupervised approach. To get a better sense of the performance compared to the best supervised methods, we measure performance on the larger Librispeech benchmark where our method achieves word error rate (WER) 5.9 on test-other. We also evaluate on six other European languages of the multilingual Librispeech benchmark [Pratap et al., 2020] and on three non-European low-resource languages.

## 2 Speech and Text Representations

Next, we describe how we build suitable speech and text representations for unsupervised learning. Good representations are essential to learning a mapping from speech to text without supervision.

### 2.1 Self-supervised Learning of Speech Audio Representations

In the first step, we learn representations of the speech audio signal using self-supervised learning. There has been a lot of recent work in this direction which has shown strong performance in extremely low-labeled data setups across a range of languages [Conneau et al., 2020] and tasks [Fan et al., 2021, Pepino et al., 2021, Wang et al., 2021].

Wav2vec 2.0 consists of a convolutional feature encoder $f : \mathcal{X} \mapsto \mathcal{Z}$ that maps a raw audio sequence $X$ to latent speech representations $z_1, \ldots, z_T$, which a Transformer $g : \mathcal{Z} \mapsto \mathcal{C}$ then turns into context representations $c_1, \ldots, c_T$ [Baevski et al., 2020b,a]. Each $z_t$ represents about 25ms of audio strided by 20ms and the Transformer architecture follows BERT [Vaswani et al., 2017, Devlin et al., 2019]. During training, latent representations are discretized to $q_1, \ldots, q_T$ with a quantization

module $\mathcal{Z} \mapsto \mathcal{Q}$ to represent the targets in the objective. Quantization uses a Gumbel softmax to choose entries from two codebooks [Jegou et al., 2011, Jang et al., 2016, Baevski et al., 2020b].

In our experiments, we use the publicly available English model pre-trained on 53k hours of Libri-Light [Kahn et al., 2020b] as well as XLSR-53 which was pre-trained on nearly 60k hours of speech audio in 53 languages [Conneau et al., 2020].

## 2.2 Pre-processing and Embedding the Audio Data

**Removing Silences.** Most datasets we use for our experiments have audio data with silences. However, these parts of the audio do not correspond to any transcription and we therefore remove silences as much as possible. We apply rVAD, an unsupervised voice activity detection (VAD) model which determines the segments in the audio data corresponding to silences, and we remove these sections [Tan et al., 2020]. We ablate this choice in Appendix C.

**Speech Audio Representations.** After silence removal, we embed the unlabeled speech audio with wav2vec 2.0 to obtain speech representations. Specifically, we use the representations of the context Transformer network $c_1, \ldots, c_T$ (§ 2.1). The context network contains 24 Transformer blocks and we denote the output of block $l$ at time-step $t$ as $c_t^l$. Our goal is to learn a model which can map from audio representations $c_t^l$ to phonemes using no supervision. However, the representations of the uppermost block of wav2vec 2.0 may not be well suited for this task. These features are trained to directly predict masked latent representations spanning 25ms of speech audio which is much shorter than the typical duration of a phoneme.

To get a better sense of this, we train supervised phoneme recognizers with a CTC loss [Graves et al., 2006] on top of the frozen representations of each of the 24 blocks of the English wav2vec 2.0 LARGE model pre-trained on Libri-Light. We then evaluate phone error rate (PER) with respect to the phonemized transcriptions of Librispeech dev-other. The classifier takes as input $c_t^l$ and contains a single softmax-normalized linear layer mapping to the phoneme inventory. Figure 2 shows that most of the first ten blocks as well as the final blocks provide very poor performance, while blocks 15-19 provide error rates below 9% PER. Block 15 achieves the best error rate of 7.5% PER. A similar insight has been used in the concurrent work of Hsu et al. [2021b]. Appendix A shows that this choice generalizes to other languages. For brevity we drop the superscript $l$ and refer to block 15 representations simply as $c_1, \ldots, c_T$.

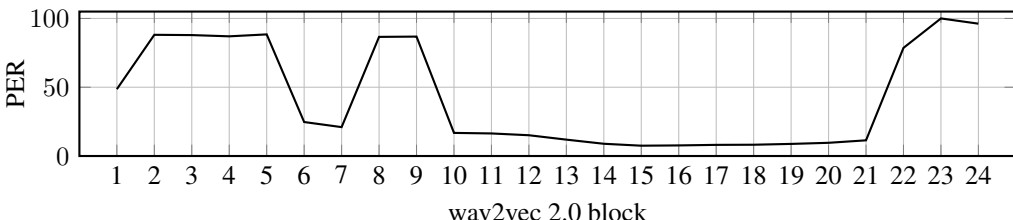

Figure 2: Supervised phoneme recognition using representations from different wav2vec 2.0 blocks on dev-other of English Librispeech. Low and high blocks do not provide good features, while as blocks 14-19 do. Block 15 performs best.

## 2.3 Segmenting the Audio Signal

Once the speech signal is embedded, we identify segments corresponding to meaningful units that can be mapped to phonemes. Segmentation has been shown to be crucial in prior work [Chung et al., 2018] since the right boundaries in the input representations make it more aligned to phonetic sequences. There has been a lot of prior work in unsupervised speech segmentation [Kamper et al., 2017a,b, Rasanen et al., 2015, Kreuk et al., 2020] but here we simply use a method based on clustering the wav2vec 2.0 speech representations $c_1, \ldots, c_T$. In a first step, we collect all the speech representations for the unlabeled speech data and perform k-means clustering to identify $K = 128$ clusters. We use the FAISS library to do fast clustering on GPUs [Johnson et al., 2019]. Next, each $c_t$ is labeled with the corresponding cluster ID $i_t \in \{1, \ldots, K\}$ and we introduce speech segment boundaries whenever the cluster ID changes.

Once the speech audio representations are segmented, we compute a 512-dimensional PCA over all speech representations output by wav2vec 2.0 for the training set. Next, we mean-pool the PCA representations for a particular segment to obtain an average representation of the segment. The PCA retains only the most important features and we found this to be effective. Segment boundaries are noisy due to the lack of supervision and we therefore found it useful to also mean-pool pairs of adjacent segment representations to increase robustness. This results in sequences of speech segment representation $S = s_1, \ldots, s_T, S \sim \mathcal{S}$ for a given utterance. Appendix B shows an illustration of the segmentation strategy on an actual example as well as a quantitative evaluation of the strategy compared to human segmented data.

## 2.4 Pre-processing the Text Data

Similar to how we segment the unlabeled speech audio data into suitable units for unsupervised learning, we do the same for the unlabeled text data. We apply two pre-processing steps to the text data: phonemization and silence token insertion.

Phonemes characterize the different sounds which distinguish words from each other, e.g., for the word *cat* there are three phonemes corresponding to the three distinct sounds in the pronunciation of the word: /K/, /AE/, /T/. We phonemize the text data because we found it easier to learn a mapping between speech audio and the different sounds of a word rather than between audio and words or letters. Phonemization converts a sequence of words $Y$ into a sequence of phonemes $P = [p_1, \cdots, p_M]$, where $p_m \in O$ and $O$ is the phoneme inventory. We use off-the-shelf tools for this step which we detail in Appendix § E.2.

The unlabeled speech audio data is pre-processed by applying unsupervised silence removal. However, this process is not always accurate and many silences in the speech audio remain. To deal with this, we enable the unsupervised model to label some segments with a phonemic silence token (SIL; § 3.1). However, the phonemized unlabeled text data does not contain any silence tokens and this may pose difficulties for adversarial learning (§ 3). We remedy this by inserting silence markers at the beginning and end of the phonemized unlabeled text data; we also randomly insert SIL between words, or groups of phonemes corresponding to words at a rate of 25%. Appendix C evaluates these choices.

# 3 Unsupervised Learning

We use adversarial training to train an unsupervised speech recognition model using the representations of the unlabeled speech audio data and the unlabeled phonemized text data [Liu et al., 2018, Chen et al., 2019]. In the following, we detail the model architecture, the training objective as well as the unsupervised cross-validation metric we developed.

## 3.1 Model Architecture

Generative adversarial networks (GAN; Goodfellow et al. 2014) train a generator network $\mathcal{G}$ and a discriminator/critic network $\mathcal{C}$ where the generator produces samples which are then judged by the discriminator. The discriminator is trained to classify whether samples are from the generator or from the real data distribution. The objective of the generator is to produce samples that are indistinguishable by the discriminator.

Concretely, $\mathcal{G}$ takes as input a sequence of $T$ segment representations $S = [s_1, \ldots, s_T]$ (§ 2.3) which are then mapped to a sequence of $M$ phonemes $\mathcal{G}(S) = [p_1, \ldots, p_M]$. The generator predicts a distribution over the phoneme set $O$ for each segment and outputs the phoneme with the highest probability. If the argmax prediction of consecutive segments result in the same phoneme, then we sample one of these segments, therefore $M \leq T$.

The phoneme set $O$ includes a silence label SIL to enable labeling silences in the speech audio as such. Without a silence label, we noticed that the model was repurposing a particular phoneme to label silences which resulted in much lower performance since it interfered with subsequent language model (LM) decoding. In the backward pass, we back-propagate through segments sampled at the generator output. We do not modify the segment representations $S$ during unsupervised training. The generator is parameterized as a single layer convolutional neural network (CNN).

The discriminator takes as input either a sequence $P^r \sim \mathcal{P}^r$ of one-hot vectors denoting phonemized text from the real data distribution $\mathcal{P}^r$ or a sequence of output distributions from the generator $\mathcal{G}(S)$. Each input vector has $|O|$ dimensions to represent the distribution over phonemes for each segment. The discriminator is also a CNN which outputs a probability indicating how likely the sample is to be from the data distribution.

## 3.2 Objective

In our setup we use the original GAN objective with a gradient penalty [Goodfellow et al., 2014, Arjovsky et al., 2017], a segment smoothness penalty and a phoneme diversity penalty:

$$\min_{\mathcal{G}} \max_{\mathcal{C}} \mathbb{E}_{P^r \sim \mathcal{P}^r} \left[ \log \mathcal{C}(P^r) \right] - \mathbb{E}_{S \sim \mathcal{S}} \left[ \log \left( 1 - \mathcal{C}(\mathcal{G}(S)) \right) \right] - \lambda \mathcal{L}_{gp} + \gamma \mathcal{L}_{sp} + \eta \mathcal{L}_{pd} \quad (1)$$

where $P^r \sim \mathcal{P}^r$ is phonemized unlabeled text, $\mathcal{G}(S)$ is the transcription output by the generator of input segment representations $S$ for some unlabeled speech audio. The first term trains the discriminator to assign high probability to real transcriptions, the second term encourages the discriminator to assign low probability to generator outputs, $\mathcal{L}_{gp}$ is a gradient penalty, $\mathcal{L}_{sp}$ is a smoothness penalty and $\mathcal{L}_{pd}$ is a phoneme diversity loss which we detail next. During training we alternate updates for the discriminator and the generator. We also alternate batches of predicted transcriptions from the generator and phonemized unlabeled text.

**Gradient penalty.** To stabilize training, we penalize the gradient norm of the discriminator with respect to the input [Gulrajani et al., 2017]. The penalty is computed for random samples $\tilde{P} \sim \tilde{\mathcal{P}}$ which are a linear combination of the activations of pairs of real and fake samples.[2]

$$\mathcal{L}_{gp} = \mathbb{E}_{\tilde{P} \sim \tilde{\mathcal{P}}} \left[ \left( \| \nabla \mathcal{C}(\tilde{P}) \| - 1 \right)^2 \right] \quad (2)$$

**Segment smoothness penalty.** The k-means segmentation of the speech audio is more granular than a typical phonemized transcription and neighboring representations are highly correlated. We therefore found it useful to add a penalty which encourages the generator to produce similar outputs for adjacent segments where $p_t \in \mathbb{R}^{|O|}$:

$$\mathcal{L}_{sp} = \sum_{(p_t, p_{t+1}) \in \mathcal{G}(S)} \| p_t - p_{t+1} \|^2 \quad (3)$$

**Phoneme diversity loss.** We also found it helpful to penalize low usage of the phoneme vocabulary by the generator on the batch level. In particular, we maximize the entropy of the averaged softmax distribution $H_{\mathcal{G}}(\mathcal{G}(S))$ of the generator over the phoneme vocabulary across a batch $B$ of utterances:

$$\mathcal{L}_{pd} = \frac{1}{|B|} \sum_{S \in B} -H_{\mathcal{G}}(\mathcal{G}(S)) \quad (4)$$

## 3.3 Unsupervised Cross-Validation Metric

Our goal is to build speech recognition models without any supervision. To this end, we developed a cross-validation metric which does not require labeled data. We use the metric for early stopping, selecting a random seed, and hyper-parameter selection ($\lambda$, $\gamma$, $\eta$).

We consider two quantities in our metric: *LM negative log-likelihood (NLL)* and *vocabulary usage*. LM-NLL serves as an indicator of fluency for a given transcription and it is measured with a language model $p_{LM}$ trained on phonemized text data (§ 2.4). Vocabulary usage is the proportion of the phoneme vocabulary being output by the model via Viterbi decoding. Measuring vocabulary usage identifies degenerate models which output fluent but trivial transcriptions.

We denote Viterbi phoneme transcriptions for a given generator configuration $\mathcal{G}$ and unlabeled speech audio $\{X_j\}_{j=1}^{N_s}$ as $\mathcal{P} = \{P_j\}_{j=1}^{N_s}$. LM-NLL is measured in the standard way over the phonemized transcriptions: $NLL_{LM}(\mathcal{P}) = \frac{1}{N_s} \sum_{j=1}^{N_s} NLL_{LM}(P_j)$ where $NLL_{LM}(P) = -\frac{1}{M} \sum_{t=1}^{M} \log p_{LM}(p_t)$

---

[2]We simply shorten longer sequence if the lengths differ.

using $p_{LM}(p_t)$ as shorthand for $p_{LM}(p_t|p_{t-1}, \ldots, p_1)$.[3] On the other hand, we use $U(\mathcal{P}) = \frac{1}{|O|} \sum_{o \in O} [o \in \mathcal{P}] \in [0, 1]$ to denote the vocabulary usage of $\mathcal{P}$.

In a first step, we generate phoneme transcriptions for different training checkpoints or hyper-parameter settings and denote the transcriptions of the configuration with the lowest vocabulary-usage adjusted NLL as $\hat{\mathcal{P}} = \arg\min_{\mathcal{P}} NLL_{LM}(\mathcal{P}) - \log U(\mathcal{P})$.[4] Next, we discard model configurations which do not satisfy the following using $\hat{\mathcal{P}}$ as the anchor:

$$NLL_{LM}(\mathcal{P}) < NLL_{LM}(\hat{\mathcal{P}}) + \log\left(\frac{U(\mathcal{P})}{U(\hat{\mathcal{P}})}\right) + \log 1.2 \tag{5}$$

The second term on the right hand side introduces a margin over the NLL of the anchor transcription $NLL_{LM}(\hat{\mathcal{P}})$ based on the vocabulary usage of $\mathcal{P}$ and $\hat{\mathcal{P}}$: If $U(\hat{\mathcal{P}})$ is much lower compared to $U(\mathcal{P})$, then we allow model configurations which produce transcriptions with higher NLL compared to $\hat{\mathcal{P}}$. However, if $U(\hat{\mathcal{P}})$ is a lot higher than $U(\mathcal{P})$, then the model configuration will not satisfy the constraint. The $\log 1.2$ factor serves as another margin allowing checkpoints with slightly worse vocabulary-usage adjusted NLL to be included.

In a final step, we take into account the length of the transcriptions: out of the configurations $\mathcal{P}'$ which satisfy the above constraint, we select the one which has the highest sum of log probability without normalizing the length:

$$\mathcal{P}^* = \arg\max_{\mathcal{P}'} \sum_{j=1}^{N_s} \sum_{t=1}^{M} \log p_{LM}(p_t^j), M = |P^j|, P^j = [p_1^j, \ldots, p_M^j] \tag{6}$$

This selects model configurations which produce phoneme sequences that score high under the language model but are not too long. Appendix D compares accuracy when developing with this metric compared to a labeled development set.

## 4   Results

### 4.1   Comparison to Supervised Speech Recognition on Librispeech

We first test our approach on Librispeech to get a sense of how unsupervised speech recognition compares to the best supervised systems trained on a large amount of labeled data. Librispeech is a standard benchmark in the speech recognition community which provides about 960 hours of transcribed read audiobooks. We use the language modeling data of Librispeech as unlabeled text data for unsupervised training. In Appendix G we show that far less unlabeled text and speech audio are sufficient to reach a similar level of performance. We experiment with the frozen representations of a wav2vec 2.0 LARGE model trained on the 53.2k hours of Libri-Light (LL-60k) which we denote as wav2vec-U LARGE. We also consider self-training over three iterations by first training an HMM on the labels generated by the GANm then fine-tuning the original wav2vec 2.0 model on the labels of the HMM for Librispeech followed by then fine-tuning on Libri-Light; Appendix F investigates alternatives.

wav2vec-U LARGE with self-training (wav2vec-U + ST) and a Transformer language model achieves WER 5.9 on test-other, the noisy test set. This shows that unsupervised speech recognition can perform remarkably well compared to the best supervised systems of the recent past on this much studied benchmark. Also, self-training is effective even when the teacher model is unsupervised as per the improvement over GAN training (wav2vec-U). Interestingly, self-training on just Librispeech, or 960 hours of unlabeled speech audio, achieves already very good performance of WER 6.4 on dev-other compared to self-training on all of Libri-Light (53.2k hours) which compares at 6.0 WER. We note that the number of parameters trained during adversarial training is very small: the generator contains only about 90k parameters for a single temporal convolution mapping to the phoneme set from frozen wav2vec 2.0 representations.

---

[3]We remove SIL labels from $\mathcal{P}$ when computing the NLL because SIL is not used in $p_{LM}$.

[4]In practice, we used language model perplexity which is equivalent to NLL after taking the log.

Table 1: WER on Librispeech dev/test sets when using 960 hours of unlabeled audio from Librispeech (LS-960) or 53.2k hours from Libri-Light (LL-60k) using representations from wav2vec 2.0 LARGE. Librispeech provides clean dev/test sets which are less challenging than the other sets. We report results for GAN training only (wav2vec-U) and with subsequent self-training (wav2vec-U + ST).

| Model | Unlabeled data | LM | dev | | test | |
|---|---|---|---|---|---|---|
| | | | clean | other | clean | other |
| **960h - Supervised learning** | | | | | | |
| DeepSpeech 2 [Amodei et al., 2016] | - | 5-gram | - | - | 5.33 | 13.25 |
| Fully Conv [Zeghidour et al., 2018] | - | ConvLM | 3.08 | 9.94 | 3.26 | 10.47 |
| TDNN+Kaldi [Xu et al., 2018] | - | 4-gram | 2.71 | 7.37 | 3.12 | 7.63 |
| SpecAugment [Park et al., 2019] | - | RNN | - | - | 2.5 | 5.8 |
| ContextNet [Han et al., 2020] | - | LSTM | 1.9 | 3.9 | 1.9 | 4.1 |
| Conformer [Gulati et al., 2020] | - | LSTM | 2.1 | 4.3 | 1.9 | 3.9 |
| **960h - Self and semi-supervised learning** | | | | | | |
| Transf. + PL [Synnaeve et al., 2020] | LL-60k | CLM+Transf. | 2.00 | 3.65 | 2.09 | 4.11 |
| IPL [Xu et al., 2020b] | LL-60k | 4-gram+Transf. | 1.85 | 3.26 | 2.10 | 4.01 |
| NST [Park et al., 2020] | LL-60k | LSTM | 1.6 | 3.4 | 1.7 | 3.4 |
| wav2vec 2.0 [Baevski et al., 2020c] | LL-60k | Transf. | 1.6 | 3.0 | 1.8 | 3.3 |
| wav2vec 2.0 + NST [Zhang et al., 2020b] | LL-60k | LSTM | 1.3 | 2.6 | 1.4 | 2.6 |
| **Unsupervised learning** | | | | | | |
| wav2vec-U LARGE | LL-60k | 4-gram | 13.3 | 15.1 | 13.8 | 18.0 |
| wav2vec-U LARGE + ST | LL-60k | 4-gram | 3.4 | 6.0 | 3.8 | 6.5 |
| | LL-60k | Transf. | 3.2 | 5.5 | 3.4 | 5.9 |

## 4.2 Comparison to Prior Unsupervised Work

Prior work on unsupervised speech recognition focused on the TIMIT benchmark. In order to perform a direct comparison to these approaches, we report results on this benchmark as well. We consider two setups to compare to previous work: in the matched setting, the unlabeled text data is simply the transcriptions of the unlabeled audio data but unpaired. In the unmatched setup, the unlabeled text data does not contain the transcriptions for the audio data which is a more realistic setting.

We measure performance on the standard Kaldi dev and test sets (core-dev/core-test) as well as a slightly larger version of the test set (all-test) to be able to compare to Liu et al. [2018] and Chen et al. [2019]. Further details of the two setups can be found in Appendix § E.1. We report performance for wav2vec-U with a 4-gram language model trained on the language modeling data of TIMIT and we also consider self-training (wav2vec-U + ST).

Table 2 shows that wav2vec-U outperforms prior unsupervised work in both the matched and unmatched settings, reducing PER on all-test in the matched setup by 57% relative compared to Chen et al. [2019]. Our method has lower performance than the best supervised methods but it performs still very well at PER 12 on core-test in the matched setup compared to PER 8.3 for the state of the art [Baevski et al., 2020c].

## 4.3 Performance on non-English languages

To get a sense of how well the method works on non-English data, we experiment on six languages of the multilingual Librispeech corpus (MLS; Pratap et al. 2020). As baseline we consider the supervised systems of Pratap et al. [2020] trained on between 2k and 161 hours of labeled data, depending on the language. For adversarial learning we use 100 hours of unlabeled audio data from MLS for every language as well as the MLS language modeling data. As input to wav2vec-U we use the representations from XLSR-53 [Conneau et al., 2020], a wav2vec 2.0 model pre-trained on 53 languages. Table 3 shows that wav2vec-U generalizes across a range of languages. Performance is lower than supervised systems but it shows the viability for other languages.

Next, we turn to three low-resource languages, Swahili, Kyrgyz, and Tatar. Swahili is an African language, Kyrgyz and Tatar are Turkic languages with only about 4.3m and 5.2m speakers, respectively.[5] We use between 1.8 hours (Kyrgyz) and 9.2 hours of unlabeled audio (Swahili), see Appendix § E.1.

---

[5] `https://en.wikipedia.org/wiki/{Kyrgyz,Tatar}_language`

Table 2: TIMIT Phoneme Error Rate (PER) in comparison to previous work for the matched and unmatched training data setups (Appendix § E.1). PER is measured on the Kaldi dev and test sets (core-dev/core-test) as well as a slightly larger version of the test set (all-test) as used by some of the prior work. ($*$) indicates experiments that do not use the standard split excluding SA utterances.

| Model | LM | core-dev | core-test | all-test |
|---|---|---|---|---|
| **Supervised learning** | | | | |
| LiGRU [Ravanelli et al., 2018] | - | - | 14.9 | - |
| LiGRU [Ravanelli et al., 2019] | - | - | 14.2 | - |
| **Self and semi-supervised learning** | | | | |
| vq-wav2vec [Baevski et al., 2020b] | - | 9.6 | 11.6 | - |
| wav2vec 2.0 [Baevski et al., 2020c] | - | 7.4 | 8.3 | - |
| **Unsupervised learning - matched setup** | | | | |
| EODM [Yeh et al., 2019] | 5-gram | - | 36.5 | - |
| GAN$*$ [Chen et al., 2019] | 9-gram | - | - | 48.6 |
| GAN + HMM$*$ [Chen et al., 2019] | 9-gram | - | - | 26.1 |
| wav2vec-U | 4-gram | 17.0 | 17.8 | 16.6 |
| wav2vec-U + ST | 4-gram | 11.3 | 12.0 | 11.3 |
| **Unsupervised learning - unmatched setup** | | | | |
| EODM [Yeh et al., 2019] | 5-gram | - | 41.6 | - |
| GAN$*$ [Chen et al., 2019] | 9-gram | - | - | 50.0 |
| GAN + HMM$*$ [Chen et al., 2019] | 9-gram | - | - | 33.1 |
| wav2vec-U$*$ | 4-gram | 21.3 | 22.3 | 24.4 |
| wav2vec-U + ST$*$ | 4-gram | 13.8 | 15.0 | 18.6 |

Table 3: WER on the Multilingual Librispeech (MLS) dataset using representations from the wav2vec 2.0 XLSR-53 model. We consider German (de), Dutch (nl), French (fr), Spanish (es), Italian (it), Portuguese (pt).

| Model | Labeled data used | LM | de | nl | fr | es | it | pt | Avg |
|---|---|---|---|---|---|---|---|---|---|
| Labeled training hours (full) | | | 2k | 1.6k | 1.1k | 918 | 247 | 161 | |
| **Supervised learning** | | | | | | | | | |
| Pratap et al. [2020] | full | 5-gram | 6.49 | 12.02 | 5.58 | 6.07 | 10.54 | 19.49 | 10.0 |
| **Unsupervised learning** | | | | | | | | | |
| wav2vec-U | 0h | 4-gram | 32.5 | 40.2 | 39.8 | 33.3 | 58.1 | 59.8 | 43.9 |
| wav2vec-U + ST | 0h | 4-gram | 11.8 | 21.4 | 14.7 | 11.3 | 26.3 | 26.3 | 18.6 |

To compare to prior work, we measure WER for Swahili and PER for Kyrgyz and Tatar. For Tatar and Kyrgyz we opted to use a reduced self-training regime for faster experimental turn-around where we only perform HMM self-training and we expect better performance with the full self-training setup (Appendix F). Table 4 and Table 5 show that wav2vec-U achieves good performance on these low-resource languages compared to previous work that utilized labeled data. We note that for Tatar and Kyrgyz we use a much smaller amount of speech audio than prior work: compared to XLSR-53 we use 1.8h unlabeled data vs 17h of labeled data for Kyrgyz and 4.6h vs. 17h for Tatar.

## 5 Related Work

This paper builds on a large body of prior work which includes semi-supervised speech recognition such as self-training [Kahn et al., 2020a, Xu et al., 2020b, Park et al., 2020]. Some of the earliest work in self-supervised learning of speech representations was was done by van den Oord et al. [2018] for phoneme recognition which was simplified in Schneider et al. [2019] who applied it to full speech recognition. Other work includes language model-style pre-training [Chung et al., 2019a] and learning fixed size representations of audio segments [Chung and Glass, 2018]. There is also work

Table 4: PER for low-resource languages, Tatar (tt) and Kyrgyz (ky).

| Model | tt | ky |
|---|---|---|
| **Supervised learning** | | |
| Fer et al. [2017] | 42.5 | 38.7 |
| m-CPC [Rivière et al., 2020] | 42.0 | 41.2 |
| XLSR-53 [Conneau et al., 2020] | 5.1 | 6.1 |
| **Unsupervised learning** | | |
| wav2vec-U | 25.7 | 24.1 |
| wav2vec-U + HMM | 13.7 | 14.9 |

Table 5: WER for Swahili from the ALFFA corpus. We compare to the supervised baseline of the ALFFA project.

| Model | sw |
|---|---|
| **Supervised learning** | |
| Besacier et al. [2015] | 27.36 |
| **Unsupervised learning** | |
| wav2vec-U | 52.6 |
| wav2vec-U + ST | 32.2 |

on quantization of the continuous speech data [Baevski et al., 2020b,a, Liu et al., 2019, van Niekerk et al., 2020, Baevski et al., 2020c, Hsu et al., 2021b] and on robustness to domain shift [Hsu et al., 2021a], multilingual pre-training [Kawakami et al., 2020, Conneau et al., 2020] as well as combining speech and vision [Harwath et al., 2020].

Learning to map speech to phonemes without supervision using adversarial learning has been explored by Liu et al. [2018] who learn a mapping matrix between segment identifiers and phonemes. However, their work still relied on data segmented into phonemes by human annotators. This has been later extended to use an automatic segmentation [Chen et al., 2019] which is iteratively refined with HMMs. However, cross validation is still performed using labeled data (personal communication with authors). We also explored HMMs to refine segmentation boundaries (Table A2) but did not find it as effective as self-training. Our work is in part inspired by aligning word embedding spaces of different languages [Mikolov et al., 2013, Artetxe et al., 2017, Conneau et al., 2018] and full unsupervised machine translation [Lample et al., 2018, Artetxe et al., 2018, Conneau and Lample, 2019].

## 6 Conclusion and Future Work

wav2vec-U is a framework which enables building speech recognition models without labeled data. It embeds and segments the speech audio with self-supervised representations from wav2vec 2.0, learns a mapping to phonemes with adversarial learning, and cross-validates hyper-parameter choices as well as early stopping with an unsupervised metric. Experiments on the standard Librispeech benchmark show performance close to the state of the art models from only a few years ago, even though these models relied on nearly 1,000 hours of labeled data.

Compared to the previous best unsupervised speech recognition approach, wav2vec-U reduces TIMIT phone error rate from 26.1 to 11.3. We also demonstrate the viability of our approach on several languages other than English, some of which are low-resource. The ability to build speech recognition models solely from unlabeled speech audio and unlabeled text drastically lowers the effort to build speech technology for many more languages of the world.

Our approach requires phonemization of the text for the language of interest. Moreover, phonemizers are not available for all languages and this presents a bottleneck. To address this, future work may develop phonemizers for more languages, explore phonemization approaches that generalize across languages, or unsupervised training with graphemic text units such as letters.

We explored a simple segmentation technique based on self-supervised representations, however, there is a large body of research on segmentation and some of these techniques may lead to improvements over our simple approach [Varadarajan et al., 2008, Zhang and Glass, 2009, Gish et al., 2009, Lee and Glass, 2012, Lee et al., 2015, Ondel et al., 2016, Kamper et al., 2017a,b, Kreuk et al., 2020]. Also, wav2vec 2.0 learns representations for fixed size units with a fixed stride, however, phonemic units are of variable size. Another direction is to learn variable sized representations during pre-training.

## Acknowledgments and Disclosure of Funding

We thank Zhouhan Lin for helping with initial explorations in this project, Tatiana Likhomanenko for helpful discussions about self-training, Da-Rong Liu for sharing details to reproduce the setup of Chen et al. [2019], Marc'Aurelio Ranzato for general helpful discussions, and Ruth Kipng'eno, Ruth Ndila Ndeto as well as Mark Mutitu for error analysis of our Swahili model.

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
