# Appendices

## A   Choosing Audio Representations

To get a sense of whether the audio representations we chose generalize (§ 2.2), we measure PER for eight different languages of the MLS dataset (§ E.1) when inputting representations of the multilingual wav2vec 2.0 XLSR-53 model. Figure A1 confirms that block 15 provides good performance across a range of languages and we will use block 15 as the representation for speech audio in all subsequent experiments.

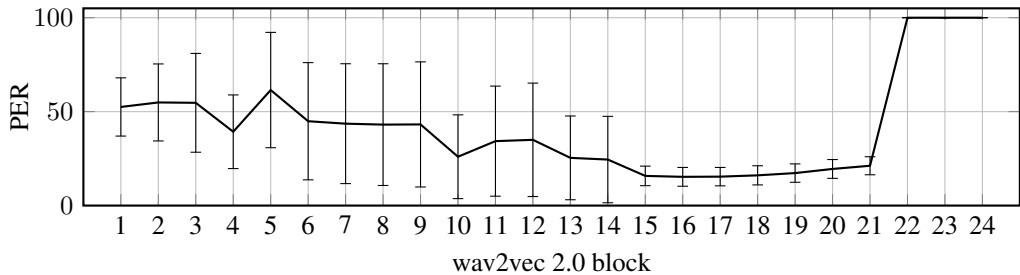

Figure A1: Supervised phoneme recognition on eight languages of the MLS dataset in terms of mean PER and standard deviation for different wav2vec 2.0 blocks to represent the raw audio (cf. Figure 2). We consider English, German, Spanish, French, Italian, Dutch, Polish and Portuguese.

## B   Segmenting the Audio Signal

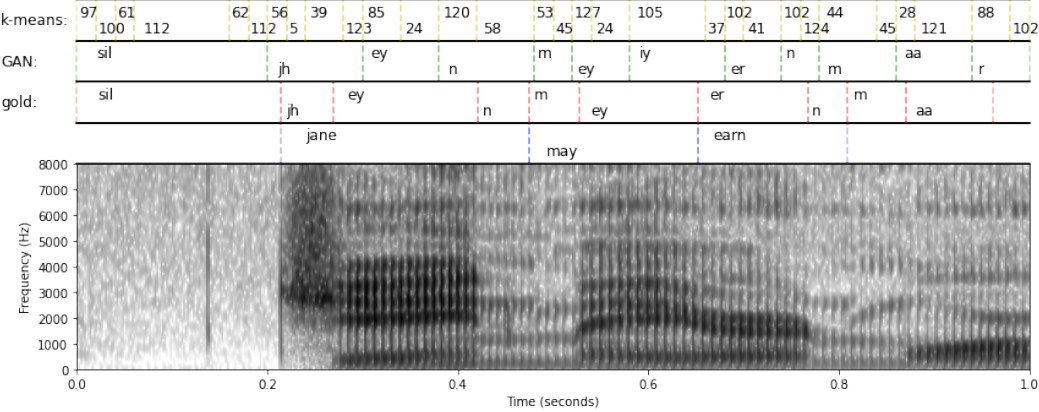

Figure A2: Example of segmenting the audio signal of the utterance *jane may earn more*. The top line shows the segmentation by the k-means segmentation method, the second line is the segmentation of Viterbi decoding with the GAN, and the third line shows the gold segmentation of human annotators from the TIMIT dataset. At the bottom, we show the corresponding spectogram, although, the input to our method is raw audio. The utterance has TIMIT ID MGLB0_SX4.

Figure A2 illustrates how the k-means segmentation strategy results in very granular units compared to the gold segments corresponding to phonemes. Based on the k-means units, the unsupervised model can then recover segments that correspond very closely to phonemic units identified by humans.

Table A1 shows that k-means clustering results in very high precision but low recall when recovering gold phoneme boundaries on TIMIT. The Viterbi outputs of our model (wav2vec-U) result in more balanced, and better, accuracy because neighboring segments with the same predicted label are combined into a larger segment.

Table A1: Quantitative evaluation of segment boundaries with respect to human labeled segment boundaries. We report precision, recall and f-measure using a 20ms tolerance.

| Method | Precision | Recall | F1 |
|---|---|---|---|
| DAVEnet + peak detection [Harwath and Glass, 2019] | .893 | .712 | .792 |
| CPC + peak detection [Kreuk et al., 2020] | .839 | .836 | .837 |
| k-means on wav2vec 2.0 features | .935 | .379 | .539 |
| wav2vec-U Viterbi prediction | .598 | .662 | .629 |

## C   Pre-processing the Audio and the Text Data

| | PER |
|---|---|
| Baseline | $21.4 \pm 1.2$ |
| w/o begin/end SIL tokens | $25.8 \pm 0.7$ |
| w/o audio silence removal | $29.3 \pm 2.0$ |

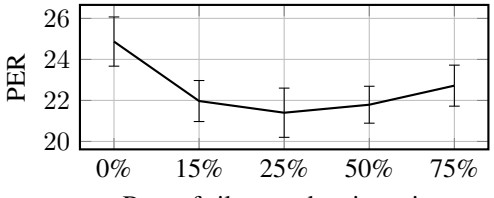

Figure A3: Unsupervised performance when augmenting the unlabeled text data with silence tokens. We add silence tokens to the unlabeled text to better resemble the speech audio which does contain silences. Silence tokens surrounding sentences and not removing silences from the audio results in better performance (left), and we show different rates of silence token insertion in the unlabeled text data (right). We report mean PER and standard deviation over 20 random seeds of unsupervised training on Librispeech dev-other.

The phonemized text data is pre-processed by adding silence tokens (§ 2.4). We empirically motivate this choice as follows: First, we add a SIL token to the beginning and the end of all phonemized unlabeled text sentences. Figure A3 (left) shows that this improves accuracy. The same table shows the detrimental effect of not removing silences from the audio data (§ 2.2). Second, we randomly insert SIL between words, or groups of phonemes corresponding to words. Figure A3 (right) shows that inserting the silence token at a rate of 0.25 yields the best end accuracy.

## D   Unsupervised Cross-Validation Metric

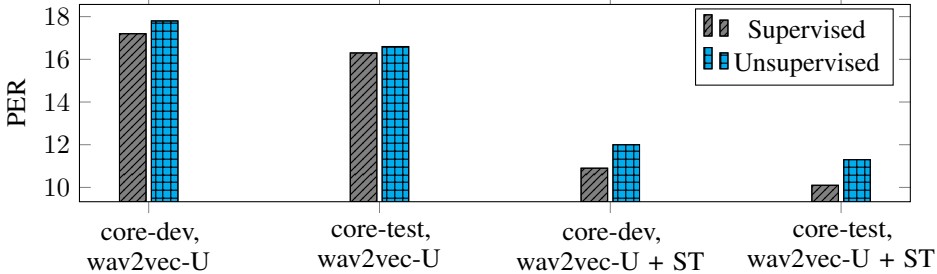

Figure A4: Effectiveness of the unsupervised cross-validation metric for model development compared to using a labeled development set (Supervised). We report PER on TIMIT core-dev/test (§ E.1) for the GAN (wav2vec-U) and with self-training (wav2vec-U + ST).

How effective is the unsupervised cross-validation metric we present (§ 3.3)? To get a sense of this, we compare the performance of cross-validation with a labeled development to cross-validation with our unsupervised metric on the TIMIT benchmark. We cross-validate GAN hyper-parameters, different checkpoints for early stopping, language model decoding hyper-parameters (§ E.4) and HMM decoding hyper-parameters for self-training (§ E.5). Figure A4 shows that the unsupervised

metric has only between 0.3-1.2 higher PER compared to using a labeled development set. This enables model development of unsupervised speech recognition systems without labeled data at only a small drop in accuracy compared to the ideal setting where labeled development data is available.

## E  Experimental Setup

### E.1  Datasets

We consider several corpora and a variety of languages to evaluate our approach. TIMIT is a small English dataset on which previous unsupervised speech recognitition work was conducted. Librispeech is a standard benchmark with nearly 1,000 hours of labeled English speech audio and MLS is a multilingual benchmark with eight European languages. In addition, we also consider non-European languages from the ALFFA and CommonVoice corpora.

**TIMIT.**  This dataset contains about five hours of audio recordings with time-aligned phonetic transcripts [Garofolo et al., 1993]. To compare to prior work, we consider two setups: the *matched* setting uses text and speech from the same set of utterances to train the model while the *unmatched* setting ensures that the unlabeled text data does not contain the transcriptions of the audio data. For the matched setup, we follow the standard train/dev/test split of TIMIT as done in Yeh et al. [2019]. This is 3,696/400/192 train/dev/test utterances which contains only SX (compact) and SI (diverse) sentences. For the unmatched setting, we follow Chen et al. [2019] by training on 3,000 speech utterances and 1,000 transcriptions from the training portion of the complete dataset split. We use the remaining 620 training utterances for validation, and test on 1,680 sentences for testing. The complete dataset split contains 4,620 training and 1,680 testing utterances, with additional SA (dialect) sentences.

**Librispeech and Libri-Light.**  The Librispeech corpus contains 960 hours of transcribed speech audio (LS-960) for training. The data is based on read English audio books. For unsupervised training, we only use the speech audio data but not the transcriptions. We use the official Librispeech language modeling data as unlabeled text data with the Libri-Light data removed [Synnaeve et al., 2020].[6] This is a large text corpus of 635m words but we show that much smaller amounts of text and speech data still result in the same performance (Appendix G). We evaluate on the standard dev-other/clean and test-clean/other sets. For development, we compute the unsupervised metric (§ 3.3) on dev-other. We also experiment with the audio data from Libri-Light (LL-60k) for which we follow the pre-processing of Kahn et al. [2020b] resulting in 53.2k hours of speech audio.

**Multilingual LibriSpeech (MLS).**  The Multilingual Librispeech dataset [Pratap et al., 2020] is a large corpus of read audiobooks from Librivox in eight languages and we experiment with the following six languages: *Dutch (du), French (fr), German (de), Italian (it), Portuguese (pt), Spanish (es)*. The latest version of this corpus contains around 50k hours including 44k hours in English. However, for unsupervised learning we only use 100 hours of speech audio for each language. As unlabeled text data, we use the LM data provided by MLS.

**ALFFA.**  We experiment with the Swahili data from the ALFFA project [Gelas et al., 2012, Abate et al., 2005, Tachbelie et al., 2014] which is read speech. There are 9.2 hours of speech audio training data and we use the language modeling data provided by ALFFA as unlabeled text data as well as newscrawl 2008-2014, 2018-2020.[7]

**CommonVoice.**  This is a multilingual corpus of read speech for 38 languages [Ardila et al., 2020]. We focus on two low-resource languages Kyrgyz (ky) and Tatar (tt) and use 1.8 hours and 4.6 hours of speech audio, respectively. As unlabeled text data for Kyrgyz we use the Kyrgyz community corpus

---

[6]https://github.com/flashlight/wav2letter/tree/master/recipes/sota/2019#
non-overlap-lm-corpus-librispeech-official-lm-corpus-excluded-the-data-from-librivox
[7]http://data.statmt.org/news-crawl/sw

2017,[8] and newscrawl 2008-2014 and 2018-2020.[9] For Tatar we use the Tatar community corpus 2017,[10] and newscrawl 2005-2011.[11] We evaluate on the dev and test split of Rivière et al. [2020].

## E.2 Phonemization

TIMIT provides time-aligned phonetic transcriptions annotated with an inventory of 60 phones adapted from the ARPAbet system, which treats silence as a phoneme. In addition, it includes a mapping from the 60 phoneme inventory to 48- and 39-phoneme inventories. Phoneme error rates are typically computed on the 39-phoneme inventory [Povey et al., 2011], which we map the phonetic transcripts to for training.

For Librispeech, we use the G2P phonemizer [Park and Kim, 2019] which uses the CMU dictionary to look up English word pronunciations, falling back to a neural network trained to output a phoneme sequence given a word. We phonemize the Librispeech LM corpus, with Librispeech and Librivox text data removed [Synnaeve et al., 2020]. We convert the full phoneme set to a reduced set containing 39 phonemes by removing the numerical stress markers from the vowels.

For other corpora, including English in the MLS dataset, we use Phonemizer which supports a large number of various languages, but is less accurate than G2P for English.[12] We disable the language-switching labels and prune phonemes that appear fewer than 1000 times in the text corpus.

## E.3 Unsupervised Training Details

Models are implemented in fairseq [Ott et al., 2019]. The generator and discriminator are optimized with Adam [Kingma and Ba, 2015] using $\beta_1 = 0.5$ and $\beta_2 = 0.98$. The discriminator has a weight decay of $1e-4$ while the generator does not use weight decay. The discriminator is trained with a learning rate of $1e-5$ and the generator with $1e-4$, which are held constant throughout the training. We train for a total of 150k steps, during which we alternate optimizing the discriminator and the generator (so each are updated 75k times in total). Each training step is performed using a batch of 160 randomly chosen samples from the unlabeled audio data and 160 randomly chosen text samples from the unlabeled text data. Training takes about 12 hours on a single V100 GPU.

The discriminator is composed of three causal convolution blocks with a hidden size of $384$ and a kernel size of 6, resulting in a receptive field size of 16 segments. The input into the discriminator is an $|O|$ dimensional vector representing the probability distribution over the phoneme vocabulary, and the output is a single logit for each time-step, indicating how likely the sample is to be from the data distribution. The first layer serves as embedding for the $|O|$ phonemes.

The generator is a single non-causal convolution with kernel size 4. The input to the generator are the segment representations $S$ of dimension 512 and the output is an $|O|$ dimensional vector. The generator contains about 90k parameters and we do not backpropagate to the segment representations. We combine subsequent generator predictions prior to feeding them into the discriminator as described in § 3.1 and apply a softmax normalization. During training, we use dropout with $p = 0.1$ to the input of the generator [Srivastava et al., 2014].

For each language, we tune the following hyper-parameters using the unsupervised cross-validation metric (§ 3.3): the gradient penalty weight $\lambda$ is selected from the range $[1.5, 2.0]$, the smoothness penalty weight $\gamma$ from $[0.5, 0.75]$, the phoneme diversity loss weight $\eta$ from $[2, 4]$, and we train 5 seeds for each configuration for a total of 40 models.

## E.4 Decoding

We wish to decode the output of either phoneme-based models, resulting from unsupervised training, or letter-based models, resulting from subsequent self-training (§ E.5) using a language model.

---

[8] https://corpora.uni-leipzig.de?corpusId=kir_community_2017

[9] http://data.statmt.org/news-crawl/ky

[10] https://corpora.uni-leipzig.de?corpusId=tat_community_2017

[11] https://corpora.uni-leipzig.de/en?corpusId=tat_news_2005-2011

[12] https://github.com/bootphon/phonemizer

To do so we build WFSTs [Mohri et al., 2002] using PyKaldi [Can et al., 2018], a Python port of Kaldi [Povey et al., 2011]. The WFST takes as input the model emissions and if we decode to words, then we use the same phonemizer with which we pre-processed the unlabeled text data to build a mapping between phonemes to words. The WFST is composed with a 4-gram language model [Heafield, 2011] pruned to keep only 4-grams occurring more than 3 times. We add self-loops that mimic CTC behavior [Zhang et al., 2020a] where blank symbols (silence for the GAN or actual blank symbol for letter models) are mapped to epsilons and consecutive predictions of the same symbol are collapsed.

During decoding, we average the predicted phoneme distributions for segments which have the same argmax prediction. We provide acoustic scale as a parameter to the Kaldi decoder and we also add a scalar $\nu$ to the blank token emission (silence for GAN models and blank for others). We tune the optimal weights for these two parameters by minimizing a quantity that measures fluency of the output as well as faithfulness to the model output. In particular we minimize the following quantity on an unlabeled development set - assuming a phoneme-based model which we wish to decode to words:

$$\sum_{j=1}^{N_s} H_{LM}(\bar{P}_j) \times \max\big(ED(\bar{P}_j, P_j), \mu\big) \tag{7}$$

where $\{P_j\}_{j=1}^{N_s}$ are the Viterbi model outputs, $\{\bar{P}_j\}_{j=1}^{N_s}$ are the word-based outputs of the WFST converted to phonemes (or simply phoneme-based outputs if decoded to phonemes), $H_{LM}(P_j)$ is the entropy of a language model, $ED$ is an edit distance such as PER for phonemes or character error rate for letter-based model models, and $\mu = 0.03$. In practice, trivial solutions may achieve very low entropy. We counteract this by replacing $H_{LM}(\bar{P}_j)$ by the average entropy of the language model training data if $H_{LM}(\bar{P}_j)$ is lower than the entropy of the training data. We tune acoustic scale in the interval $[0, 8]$, and $\nu$ in $[-3, 8]$.

For Librispeech experiments, we also decode with a Transformer language model [Baevski and Auli, 2018] trained on the Librispeech LM corpus using the beam search decoder of Pratap et al. [2019]. The Transformer LM is identical to Synnaeve et al. [2020] and contains 20 blocks, model dimension 1,280, inner dimension 6,144 and 16 attention heads. We tune acoustic scale with a beam of 50 and test performance is measured with beam 500.

### E.5   Self-Training

For self-training, we perform two iterations: first, we pseudo-label the training data with the unsupervised GAN model and train an HMM on the pseudo-labels. Second, we relabel the training data with the HMM and then fine-tune the original wav2vec 2.0 model using the HMM pseudo-labels with a CTC loss [Graves et al., 2006]. HMM models use phonemes as output, while wav2vec 2.0 models use letters. Both are decoded using WFST decoders into words. wav2vec 2.0 self-training for Librispeech uses the same fine-tuning parameters as the original wav2vec 2.0 model fine-tuned on 100 hours of Librispeech data, but we reduce masking probability to 0.25, reduce the batch size to 800k frames and train on 8 V100 GPUs for 80k updates.[13] We use the last checkpoint instead of early stopping. For TIMIT self-training, we use the one hour fine-tuning parameters of the original wav2vec 2.0 model.[14] This performs 13k updates on 4 V100 GPUs.

## F   Self-training Strategies

The self-training strategy we use is as follows: once the GAN is trained, we use it together with a language model to pseudo-label the unlabeled audio, then we train an HMM on the labels and repeat pseudo-labeling with the HMM in order to fine-tune the wav2vec 2.0 model whose representation were originally fed to the GAN. Finally, we use the fine-tuned wav2vec 2.0 model to decode the test set.

---

[13]https://github.com/pytorch/fairseq/blob/master/examples/wav2vec/config/finetuning/vox_100h.yaml

[14]https://github.com/pytorch/fairseq/blob/master/examples/wav2vec/config/finetuning/vox_1h.yaml

Table A2: PER on TIMIT for various self-training strategies. We compare the performance of just the GAN output (wav2vec-U) to one or two iterations of subsequent self-training with an HMM. We contrast this to using the HMM for re-segmenting the audio data as done in prior work [Chen et al., 2019]. We also consider self-training based on fine-tuning the original wav2vec 2.0 model (fine-tune) in or two self-training iterations [Xu et al., 2020a] as well as a combination of HMM and fine-tuning-based self-training.

| Model | LM | core-dev | core-test | all-test |
|---|---|---|---|---|
| wav2vec-U | 4-gram | 17.0 | 17.8 | 16.6 |
| + HMM | 4-gram | 13.7 | 14.6 | 13.5 |
| + HMM + HMM | 4-gram | 13.3 | 14.1 | 13.4 |
| + HMM resegment + GAN | 4-gram | 13.6 | 14.4 | 13.8 |
| + fine-tune | 4-gram | 12.0 | 12.7 | 12.1 |
| + fine-tune | - | 12.1 | 12.8 | 12.0 |
| + fine-tune + fine-tune | - | 12.0 | 12.7 | 12.0 |
| + HMM + fine-tune | - | 11.3 | 11.9 | 11.3 |
| + HMM + fine-tune | 4-gram | 11.3 | 12.0 | 11.3 |

Table A2 shows that fine-tuning with an HMM (wav2vec-U + HMM) leads to substantial improvements, however, a second iteration of HMM self-training leads to much smaller additional gains (wav2vec-U + HMM + HMM). Using the HMM to re-segment the speech audio followed by repeated GAN training (wav2vec-U + HMM resegment + GAN), similar to Chen et al. [2019], does not improve performance over just HMM self-training.

Another option is to directly fine-tune wav2vec 2.0 on the labels assigned by the GAN model (wav2vec-U + fine-tune) and this performs very well. However, another round of self-training based on a fine-tuned wav2vec 2.0 model does not improve performance (wav2vec 2.0 + fine-tune + fine-tune). We believe that this is due to overfitting since the fine-tuned model has over 300m parameters. This is in line with recent observations about overfitting in self-training for speech recognition [Likhomanenko et al., 2021].

The HMM is less likely to overfit in the way the LARGE wav2vec 2.0 model does. We therefore found it effective to perform a single round of HMM self-training followed by wav2vec 2.0 fine-tuning (wav2vec-U + HMM + fine-tune). After two rounds of self-training, we do not require a language model anymore for this benchmark which is likely because the language model has been distilled into the model to a large degree.

## G    Amount of Unlabeled Data Needed

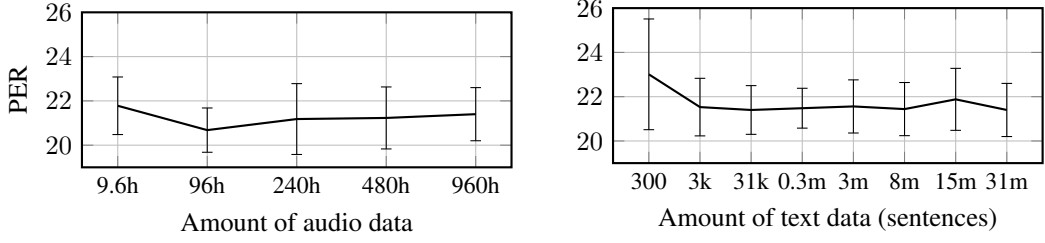

Figure A5: Effect of the amount of unlabeled audio data (left) and text data (right) on unsupervised training in terms of PER on Librispeech dev-other.

For the experiments on Librispeech we used large amounts of unlabeled speech audio and text data for adversarial learning (960 hours of unlabeled speech audio and nearly 31m sentences of text data). For TIMIT we used much less data, only about 3.15h of speech audio and 140k phonemes of text data for the matched setup. Next, we perform controlled experiments on Librispeech to get a sense of how much data is sufficient to achieve good performance.

Figure A5 (left) shows that 9.6h of speech audio data still achieves excellent performance. Similarly, Figure A5 (right) shows that only about 3,000 sentences of text data are sufficient to achieve a similar level of accuracy as using all of the text data.

# H Hyperparameter Ablations

Table A3: Ablation of various data settings, pre-processing steps, cluster sizes, PCA sizes and using the full phoneme set.

| Ablation | mean PER $\pm$ std | %-converged (PER $<$ 40) |
|---|---|---|
| Baseline | 21.4 $\pm$ 1.2 | 100% |
| 9.6h audio, 3k text | 21.2 $\pm$ 1.1 | 100% |
| 96h audio, 3k text | 21.1 $\pm$ 1.3 | 95% |
| w/o clustering, pca, mean pool | - | 0% |
| w/o clustering | - | 0% |
| w/o 2nd stage mean pool | - | 0% |
| w/o PCA | - | 0% |
| 64 clusters | 23.1 $\pm$ 0.7 | 100% |
| 256 clusters | 22.3 $\pm$ 1.1 | 100% |
| 256 PCA | 21.6 $\pm$ 1.1 | 100% |
| 768 PCA | 28.0 $\pm$ 1.5 | 90% |
| use full phone set | 23.51 $\pm$ 1.3 | 100% |