# OpenReview forum: "Unsupervised Speech Recognition"
_NeurIPS.cc/2021/Conference — NeurIPS 2021 Oral_

### Official Review · Reviewer_A9gn · 2021-07-12

**Rating:** 7
**Confidence:** 3

**Summary:**

This paper presents an unsupervised training method for speech recognition, using unlabeled audio and corresponding text. The method makes use of adversarial training, and previously proposed unsupervised learned audio representations with wav2vec 2.0.


**Limitations And Societal Impact:**

None that I know.

**Main Review:**

This work builds on a lot of existing work but extends and varies it.

Many experiments have been done including ablation studies.

How to get from phonemes (or what else?) to chars/words in the Librispeech model?
What exactly is the final model to perform speech recognition? How is recognition done?
Does it use a predefined phoneme set including the lexicon (phone to word mapping) in recognition?

Why does it need silence removal if anyway some sil is kept, and sil is explicitly modeled?

Why is PCA done here? Why does it need the segmentation?

The results look very nice, and demonstrate that this is an important contribution to this area.


**Time Spent Reviewing:**

2

---

> ### Author Response · Authors · 2021-08-10
> **Response to reviewer A9gn**
>
> We thank the reviewer for their comments!
>
> * Mapping from phonemes to chars/words: please see Appendix E.4 for details. Broadly, we use Kaldi WFST models to map from phones to words. This requires a lexicon that provides a phonemic transcription for each word. We generate this lexicon using publicly available phonemizers such as G2P-en, espeak and espeak-ng.
> * Final model performing speech recognition: the first model (GAN) learns to transcribe a phonemic sequence from raw audio.  We then perform self-training (see Appendix E.4) using a Kaldi HMM model or the original wav2vec 2.0 and the transcriptions of the GAN.
> * Silence removal: the silence token (SIL) is only kept for the TIMIT dataset where it is part of the official transcriptions. For other datasets, we remove silences with an unsupervised VAD tool (rVAD). However, this tool is imperfect and does not remove all of the silences. To better map audio to phonemic sequences, we insert silences at the beginning and end of each utterance, as well as randomly at word boundaries at a pre-defined rate (see Appendix C). This is only used for the initial GAN training, and we discard silences for subsequent stages as they never appear in the transcripts.
> * We empirically found that the generator works best when only the most important information from wav2vec features is kept, and PCA is a means to do so. There is a significant degradation in convergence rate when using raw wav2vec 2.0 features without PCA dimensionality reduction.
> * Segmentation is needed to better align the number of timesteps produced by the wav2vec 2.0 model to the number of phones generally seen in the transcription

---

### Official Review · Reviewer_kw26 · 2021-07-16

**Rating:** 8
**Confidence:** 4

**Summary:**

This paper builds upon previous work on unsupervised speech recognition. This work introduces several improvements which are crucial to achieve impressive performance on a variety of datasets. These improvements are: using a new strategy for cluster segmentation for better alignment with the phonemes, matching the number of silences for the GAN generator. Additionally, this work uses a lightweight GAN. A distinguishing features of this work comparing to prior ones is that it proposes and uses an unsupervised metric for the development set for the model selection.

The experiments include several popular datasets such as LibriSpeech, TIMIT, as well as low resource language datasets for Tatar, Kyrgyz, Swahili. The proposed method shows performance comparable to some supervised methods and superior to previous unsupervised ones.

**Limitations And Societal Impact:**

This work has a positive social impact as it helps transcription of low-resource languages.

**Main Review:**


# Originality

The paper builds upon the prior works, but it introduces several distinguishing features that are crucial for the results.

# Quality

The model described thoroughly and the architecture choices motivated well. The experiments are well motivated too. The paper shows that the experimentation is conducted rigorously. Specifically, the setup for all the datasets is described fully and should be possible to replicate. It is especially valuable that the paper conducts experiments on the low resource languages. This is where the unsupervised approach can make the difference.

# Clarity

The paper is very well written. Every section has a clear motivation. Then, the writing is simple and concise, but conveys all the information.

The only thing I found a bit lacking is that self-training (specifically, wav2vec-U + ST) is just mentioned briefly. This model seems to perform best and it makes sense to dedicate some space to describe it in details.

# Significance

In my opinion, this paper is highly significant for both speech recognition and unsupervised training communities. It is known that the unsupervised training for structured prediction (sequence-to-sequence) is hard. The speech recognition is a hard and competitive task as well. This paper can make a significant impact in both areas.

EDIT: Rebuttal acknowledged

**Time Spent Reviewing:**

5

---

> ### Author Response · Authors · 2021-08-10
> **Response to reviewer kw26**
>
> Thank you for your comments!
>
> Please see Appendix E.5 for details on the self-training procedure. It closely follows prior work [1,2]. We decided to dedicate more of the main paper to the novel parts of our work. The sole difference to classical self-training is our teacher model: in the first round, it is simply wav2vec-U.
>
> [1] Kahn et al. https://arxiv.org/abs/1909.09116 \
> [2] Xu et al. https://ieeexplore.ieee.org/abstract/document/9414641

---

### Official Review · Reviewer_zM92 · 2021-07-16

**Rating:** 8
**Confidence:** 4

**Summary:**

The authors introduces a fully unsupervised automatic speech recognition framework built upon the Wav2Vec 2.0 representations and a GAN generator/discriminator training process that, amongst others, performs on a par with previous state-of-the-art ASR models.

**Limitations And Societal Impact:**

I think that the major societal impact is written into the article.

**Main Review:**

Thank you for a really nice and informative read. The paper is well-structured and the appendix is supporting the claims and choices such that it seems reproducible. I believe this to be a strong contribution to the research community.

Some comments:
I'm not sure that I can make sense of the "Removing Silences" paragraph in the paper in cohesion with the findings in Appendix C?

In the "Speech Audio Representations" paragraph it would be informative to run a more in depth analysis on the features. E.g., "ON SCALING CONTRASTIVE REPRESENTATIONS FOR LOW-RESOURCE SPEECH RECOGNITION" by Borgholt et al. runs an interesting analysis in their Figure 3. Figure 2 in this paper seems rather arbitrary and needs investigation.

The paper would be significantly stronger from more datasets. E.g., what about more noisy and conversational datasets? How does the model perform in those settings?

Have you tried to set up a semi-supervised or self-supervised variant of the framework? It would be very interesting to see how the model framework would improve.


**Time Spent Reviewing:**

1.5

---

> ### Author Response · Authors · 2021-08-10
> **Response to reviewer zM92**
>
> We thank the reviewer for their suggestions!
>
> * The motivation for the analysis presented in Figure 2 is solely to identify a block that has features most closely related to phonemes. This is because we are interested in mapping wav2vec features to phone targets. While one can do many types of analysis to discover this, we chose a straightforward linear classification approach that is widely used in the computer vision literature (MoCo [1], SimCLR [2], etc)
> * While we did most of our investigations on the English Librispeech and TIMIT datasets, we also show that our approach works on a variety of different languages and datasets: see Tables 3, 4 and 5.  Because the wav2vec models were pretrained in the read-speech domain, we applied w2v-U on similar established benchmarks, also based on read-speech. We agree that it would be interesting to either (i) build representations on large amounts of noisy conversational datasets (although less easy to find publicly) and apply wav2vec-U in that domain, or (ii) study cross-domain transfer as for example in Robust wav2vec [3]. We leave this study for future work.
> * Our framework is based on features from self-supervised learning (wav2vec 2.0) and then applies classical self-training (semi-supervised learning) to improve upon the GAN training. Self-training clearly improves the results. We haven’t tried using a GAN objective as an auxiliary loss in a supervised setting.
>
> [1] He, et al. https://arxiv.org/abs/1911.05722 \
> [2] Chen, et al. https://arxiv.org/pdf/2002.05709.pdf \
> [3] Hsu, et al. https://arxiv.org/pdf/2104.01027.pdf

---

> > ### Comment · Reviewer_zM92 · 2021-08-28
> > **Thank you**
> >
> > Thank you very much for the response. My recommendation for the paper still stands.

---

### Decision · Program_Chairs · 2021-09-27

**Decision:**

Accept (Oral)

**Comment:**

This work was recommended for acceptance by all reviewers. They praised the thoroughness and replicability of the experimental setup, as well as the significance of the results. This work has the potential to strongly impact speech recognition for low-resource languages, and constitutes a step change in our ability to build practically usable speech recognition systems.